# Headache in the Workplace: Analysis of Factors Influencing Headaches in Terms of Productivity and Health

**DOI:** 10.3390/ijerph19063712

**Published:** 2022-03-21

**Authors:** Nicola Magnavita

**Affiliations:** 1Postgraduate School of Occupational Health, Università Cattolica del Sacro Cuore, Largo F. Vito 1, 00168 Roma, Italy; nicolamagnavita@gmail.com; 2Department of Science of Woman, Child and Public Health, A. Gemelli Foundation IRCCS, Largo A. Gemelli 8, 00168 Roma, Italy

**Keywords:** anxiety, depression, metabolic syndrome, leadership, effort, reward, sleep, workplace health promotion, medical surveillance, headache disorders

## Abstract

Headache is a very common condition that can have a significant impact on work. This study aimed to assess the prevalence of headaches and their impact on a sample of 1076 workers from 18 small companies operating in different sectors. The workers who volunteered to participate were asked to fill in the Headache Impact Test-6 (HIT-6) and answer questions designed to assess stressful and traumatic factors potentially associated with headaches. The volunteers subsequently underwent a medical examination and tests for diagnosing metabolic syndrome. Out of the 1044 workers who completed the questionnaire (participation rate = 97%), 509 (48.8%) reported suffering from headaches. In a multivariate logistic regression model, female gender, recent bereavement, intrusive leadership, and sleep problems were significantly associated with headaches. In univariate logistic regression models, headache intensity was associated with an increased risk of anxiety (OR 1.10; CI95% 1.09; 1.12) and depression (OR 1.09; CI95% 1.08; 1.11). Headache impact was also associated with the risk of metabolic syndrome (OR 1.02; CI95% 1.00, 1.04), obesity (OR 1.02, CI95% 1.01; 1.03), and reduced HDL cholesterol (OR 1.03; CI95% 1.01; 1.04). The impact of headache calls for intervention in the workplace not only to promote a prompt diagnosis of the different forms of headaches but also to improve work organization, leadership style, and the quality of sleep.

## 1. Introduction

Headache is a very common disorder. According to the Global Burden of Disease study, headache disorders are the second leading cause of years lived with disability worldwide and the leading cause among people under 50 years of age [1]. Headache disorders affect approximately 90% of people during their lifetime. Studies conducted to evaluate headache prevalence have revealed that among the adult population, tension-type headache affects 38% of the population, while migraine affects 12% and is the most disabling [2]. On the basis of these data, the World Health Organization introduced the Global Campaign to Reduce the Burden of Headache with the ultimate aim of reducing the burden of headache worldwide [3].

There are several types of headache disorders. The International Classification of Headache Disorders-III (ICHD-III) divides them into three categories: primary headaches (e.g., migraine, tension-type headache, and trigeminal autonomic cephalalgias); secondary headaches (e.g., caused by trauma or injury to the head and/or neck, cranial and/or cervical vascular disorder, non-vascular intracranial disorder, a drug or its withdrawal, infection, disorder of homoeostasis, facial or cervical structure disorders, and psychiatric conditions), and lastly other headaches (e.g., painful cranial neuropathies and other facial pain) [4].

Many work-related problems have been attributed to headache. Environmental causes include exposure to toxic substances such as lead [5] and tobacco [6], micro-pollutants [7], and ergonomic problems [8]. An Italian survey reported that headache was one of the most frequent symptoms in office workers: 5.7% of male and 9.3% of female office employees declared that it had affected them every week during the previous three months and was allegedly related to lighting, heat, drafts, odors, and poor air quality in the occupational environment [9]. Night work may be a risk factor. Studies have indicated that undefined headache and migraine are more prevalent among shift workers than day workers [10] and are often associated with sleep problems [11]. Moreover, shift work appears to be associated with chronic migraine and greater headache-related disability [12]. Occupational stress and the use of computer monitors are the most common risks described by workers with headache disorders [13]. Dostálová et al. [14] found video terminal work to be associated with headaches, especially in workers with visual problems. The quality of workplace lighting is important, and there is evidence that poor lighting may be associated with headaches [15]. A high frequency of headaches has been observed among healthcare workers during the COVID-19 pandemic, which could be caused by acute infection [16,17,18] or its long-term consequences [19,20], but also possibly due to the use of face masks [21,22] and excessive psychosocial stress caused by the pandemic [23,24,25]. Psychosocial factors seem to play a significant role in the onset of headaches. Studies have shown that an excessive psychosocial burden resulting from work demand, insufficient control over work, and dissatisfaction with uninteresting work is associated with headache [26]. Low skill discretion and low decision authority [27], role conflict, poor social climate, bullying/harassment, and effort-reward imbalance have consistently been associated with higher odds of headache in a number of studies [28,29,30]. Workplace injustice [31] and leadership styles can also affect workers’ headaches. While abusive supervision has been associated with an increased risk of headache, transformational leadership has been associated with a decrease in the same risk [32]. Anxiety and depression have also been significantly associated with headache disorders [33,34].

Headache disorders are important causes of disability worldwide [35]. Disability due to headache not only causes a significant reduction in the quality of life for affected individuals, it also creates an economic burden on society owing to a long-term decrease in productivity resulting from headache-related absenteeism and presenteeism [36,37,38]. 

Although headache is an important and widespread problem, few studies have attempted to quantify the prevalence of headache and its impact on productivity in the workplace. The aim of this study was to evaluate the frequency of headache, the factors associated with its presence and its impact on productivity. We also studied the association of headache with mental and cardiovascular health.

## 2. Materials and Methods

### 2.1. Population

In Italy, workers exposed to occupational hazards undergo an annual medical examination in the workplace. This health promotion program was implemented in 18 small companies operating in sectors principally related to health, social healthcare, trade, and energy. The various occupational tasks in these companies included office work, plumbing, electrical work, fuel distribution, food sale, personal assistance, and cleaning. While waiting for their medical examinations, workers were invited to fill in a questionnaire. Participation in the survey was voluntary. During the subsequent medical examination, the occupational physician checked the questionnaire and was able to delve more deeply into anamnestic data and, if necessary, refer the worker to National Health Service facilities for diagnostic tests or treatment.

The study received the approval of the Ethics Committee of the Università Cattolica del Sacro Cuore (project number 3008, approved 5 June 2020).

Data have been deposited in a publicly available database.

### 2.2. Questionnaire

The workers who agreed to participate in our survey were administered a questionnaire designed to assess the impact of headache. They were also given additional questionnaires to evaluate factors potentially associated with the disorder.

The Headache Impact Test-6 (HIT-6) [39] is composed of six items scored on a 5-point Likert scale (never = 6, rarely = 8, sometimes = 10, very often = 11 and always = 13). Respondents are classified according to four levels of headache impact: little or no impact (<50), some impact (50–55), substantial impact (56–59), and very severe impact (≥60). In order to ascertain the overall quality of life, the HIT-6 explores two domains regarding the pain itself (in terms of severity and functional decline) and the psychological impact of headache (vitality, cognitive, social and role functioning) [40]. This questionnaire has proved to be a reliable and valid tool [41,42,43,44,45]. In this study, the reliability of the questionnaire (Cronbach’s alpha) was 0.937.

In addition to questions concerning socio-demographic elements, participants were asked to indicate if they were working the night shift, if they had suffered injuries, road accidents or deaths in the family in the previous 12 months, and if they had suffered violence at work. The latter was tested using the Violent Incident Form (VIF), which investigated physical assault, threats and harassment that had occurred at work in the previous year [46]. These stressors were investigated because previous studies had demonstrated that these factors are important for workers’ health [47].

Intrusive leadership (IL), i.e., the tendency of the supervisor to cross over into the personal and family sphere of the employee, was investigated with three items of the Italian version [48] of Schmidt’s Toxic Leadership Scale [49] (e.g., “Does your supervisor invade the privacy of employees?”). Participants were asked to report the frequency with which they experienced the intrusive leadership of their supervisors on a 5-point Likert-type scale ranging from 1 = “never” to 5 = “always”. The reliability of the questionnaire in this study was 0.769.

The frequency of requests to perform occupational activities outside working hours was measured using the Off-Working Hours Technological Assisted Job Demand (OFF-TAJD) questionnaire [50] composed of 3 items scored on a 5-point Likert scale, ranging from 1 = never to 5 = always. Cronbach’s alpha in this survey was 0–905. The IL and OFF-TAJD were studied because they had been associated with mental well-being in previous studies [49].

Occupational stress was measured using a short Italian version [51] of the Effort/Reward Imbalance Questionnaire [52] based on Siegrist’s model [53] since workplace exposure to a recurrent lack of reciprocity between efforts spent and rewards received can increase the risk of incident stress-related disorders. The short version of the questionnaire includes three questions for the effort variable and seven for the reward variable. All items have graded responses on a 4-point Likert scale, so the resulting sub-scales are, respectively, between 3 and 12 (effort) and between 7 and 28 (reward). The weighted relationship between the two variables, effort/reward imbalance (ERI), indicates a state of distress if values are greater than one. In this study, the test score reliability coefficient (Cronbach’s alpha) for the ERI effort sub-scale was 0.855, while Cronbach’s alpha for the reward sub-scale was 0.697.

The quality of sleep was assessed using the Italian version [54] of the Pittsburgh Sleep Quality Index (PSQI) [55], consisting of 18 questions. A score of 5 or more indicates poor sleep quality (bad sleeper). Cronbach’s alpha in this study was 0.892.

Anxiety and depression were assessed using the Goldberg Anxiety and Depression Scale (GADS), Italian version [56], an 18-item self-report list of symptoms developed specifically to evaluate the probability of the onset of anxiety or depression. Each subscale consists of nine binary questions, to which a point is assigned for each affirmative answer. A score of 5 or more on the anxiety subscale, or 2 or more on the depression subscale, indicates suspected clinically evident anxiety or depression [57]. In this survey, Cronbach’s alpha for the GADS anxiety sub-scale was 0.851, and 0.792 for depression.

Components of metabolic syndrome were defined according to the International Diabetes Federation (IDF) [58], the National Cholesterol Education Program Expert Panel on Detection Evaluation and Treatment of High Cholesterol in Adults (NCEP/ATP III) [59], and the American Association of Clinical Endocrinologists (AACE) [60]. Overweight was defined as a BMI ≥ 25 kg/m^2^, or a waist circumference of ≥94 cm for men and ≥80 cm for women (obesity = BMI > 30, waist ≥102 cm for men, ≥88 cm for women), while a serum triglyceride level > 150 mg/dL (1.7 mmol/L) was classified as hypertriglyceridemia. A low level of high-density lipoprotein (HDL) serum cholesterol was defined as a serum HDL cholesterol < 40 mg/dL (1.03 mmol/L) in men and <50 mg/dL in women. A systolic blood pressure > 130 mmHg, and/or a diastolic blood pressure > 85 mmHg, or drug treatment for hypertension were classified as high blood pressure, while a plasma glucose level > 100 mg/dL (5.6 mmol/L) NCEP > 110 mg/dL (6.1 mmol/L), or the presence of hypoglycemic drug treatment were classified as high fasting glucose. The presence of three or more abnormalities in the aforementioned components was deemed to constitute metabolic syndrome (MetS) [61].

### 2.3. Statistics

Socio-demographic features were analyzed using frequency or statistical distribution for categorical and continuous variables, respectively. Cronbach’s alpha was used to verify the reliability of the variables obtained from the questionnaire. Variable means were compared using the Student’s *t*-test for parametric variables and the Wilcoxon Mann–Whitney U test for non-parametric variables.

Logistic regression analysis was used to evaluate the association between socio-demographic variables or occupational stressors and the presence of headaches. The estimated effect was presented in terms of odds ratio (OR) and 95% confidence intervals. Each of the variables was initially posited as an independent variable in univariate models in which the presence of a headache was the dependent variable. Subsequently, a multivariate logistic regression model was constructed by placing as predictors all the variables that revealed a significant increase or reduction in the odds ratio.

Linear regression with stepwise backward selection was used to determine which of the variables of interest had the greatest effect on the impact of headache. An adjusted coefficient of determination (adjusted R^2^) was then used to evaluate the resulting models.

Furthermore, logistic regression was used to evaluate the association of headache with mental health problems (anxiety and depression) and metabolic disorders (hypertension, obesity, dyslipidemia, hyperglycemia).

Statistical analyses were performed using the IBM Statistical Package for Social Sciences, SPSS, version 26.0 (IBM, Armonk, NY, USA). We applied the significance criterion for a 2-tailed *p*-value ≤ 0.05.

## 3. Results

### 3.1. Headache Prevalence and Associated Factors

During the observation period, 1076 individuals were invited to participate in the survey. A total of 1044 workers (mean age 45.78 ± 11.33 years), most of whom were female (684, 65.5%), completed the questionnaire (participation rate = 97%). Of the 509 persons (48.8%) who reported suffering from headache, 141 (27.7%) were male and 368 (72.3%) female.

Logistic regression was used to study the association of headaches with demographic characteristics (gender, age) and with the most frequent occupational stress factors. A significantly increased risk of headaches was found for the female gender, while age was not significantly associated with the disorder. In univariate models, the risk of headache was significantly higher in individuals who had suffered violence at work, domestic accidents, road accidents or bereavement in the previous year. A very significant increase in risk was also observed in workers who had an intrusive leader, were required to work overtime, exerted high effort or had low work-related rewards. Although the findings failed to associate night work with a significant increase in risk, poor quality of sleep was associated with a very significant increase in the risk of headache.

In a multivariate model in which all variables associated with increased risk in the univariate models were simultaneously entered as independent variables, and the presence of headache was the dependent variable, only female gender, poor quality sleep, and intrusive leadership were significantly associated with the onset of headaches. Among the trauma, only bereavement was associated with headache in the multivariate model (Table 1).

### 3.2. Headache Impact and Associated Factors

The impact of headache, measured using the HIT-6, scored an average of 44.34 ± 10.11 points. It was severe (>55 points) in 38.7% of the participants suffering from this disorder, thus corresponding to 18.9% of the workforce. 

In an analysis involving only the workers who reported suffering from headaches, a comparison of the 2 genders showed that headache impact on productivity was significantly higher in females (*n* = 368; age 46.8 ± 10.7; HIT-6 = 53.3 ± 8.5) than in males (*n* = 141, age = 43.7 ± 11.1; HIT-6= 49.6 ± 8.50), The disparity, evaluated by means of the Mann–Whitney U test, was highly significant (*p* < 0.001). 

To ascertain which of the factors associated with the onset of headaches had the greatest impact on productivity, we applied linear regression analysis with stepwise selection. This analysis confirmed that the impact of headache was mainly influenced by poor sleep quality. Sleep quality alone affected one-fifth of the variability in the impact of headache. Rewards received for work performed were able to have a moderating effect. A model that included intrusive leadership and the rewards received for work, as well as the quality of sleep, accounted for about a quarter of the variability of headache (Table 2).

According to the GADS score, a psychiatric examination would have diagnosed anxiety in 426 workers (40.8%) and depression in 551 individuals (52.8%).

In univariate logistic regression models, headache level was associated with an increased risk of anxiety (OR 1.10; CI95% 1.09; 1.12) and depression (OR 1.09; CI95% 1.08; 1.11). Headache impact was also a predictor of the risk of metabolic syndrome (OR 1.02; CI95% 1.00, 1.04), obesity (OR 1.02, CI95% 1.01; 1.03), and reduced HDL cholesterol (OR 1.03; CI95% 1.01; 1.04). (Table 3).

## 4. Discussion

This study has shown that headaches affect nearly half of workers who undergo annual health surveillance in the workplace and severely interfere with the work of one in five. Headache is more frequent and more intense in women and can be influenced by numerous work-related stressors such as violence at work, traffic accidents, injuries, trauma, excessive effort, poor rewards, intrusive leadership, and off-time work. Poor sleep is strongly associated with headache, and the impact on work is made worse by intrusive leadership and lack of rewards for work. This suggests that management style plays a vital role in determining the negative effects of headache on productivity.

In this study, headache impact, measured by the HIT-6 score, was a significant predictor of mental and metabolic disorders. Studies on headache prevalence have reported that tension-type headaches are the most common form, while 12% are affected by migraine [2]. Migraines are frequently under-diagnosed and misreported [62]. The workers we studied had both types of headaches. 

In the literature, depression and anxiety are among the most common comorbidities of primary headaches [63]. A population-based study demonstrated that anxiety and depression measured by the GADS questionnaire were more frequent in subjects suffering from headaches than in controls. The HIT-6 score was significantly higher among individuals affected by anxiety than non-anxious participants [64]. This study confirms that the same occurs in the working population. Anxiety and the severity of headaches are closely associated with presenteeism and workers’ productivity [65]. Headache is one of the most common causes of health-related productivity loss [66]. Migraine is associated with cardiovascular disease, psychiatric disease, and sleep disorders [67]. A meta-analysis of 16 cohort studies involving more than one million patients indicated that migraine is associated with an increased risk of major adverse cardiovascular and cerebrovascular events, stroke, and myocardial infarction [68]. The findings of this study confirmed the association between headaches and cardiovascular risk factors in workers. In particular, the association between headache and obesity was in agreement with pathophysiological studies demonstrating that both obesity and headache could be linked to mechanisms such as inflammation and irregular hypothalamic function. This leads us to believe that dietary strategies for weight loss may be able to ameliorate headache/migraine [69]. Epidemiological and clinical considerations demonstrate that obesity increases morbidity in migraine and headache, whereas weight loss can improve headache morbidity [70]. The association between migraine and dyslipidemia has been observed in case-control [71] and population studies [72].

Despite the impact of headache on output, it has been observed that there is a lack of reliable data on headache-related work factors associated with productivity [73]. This study aims to reduce this information gap. In this research, headaches were associated with numerous stressors and sleep problems. Some of these associations have long been demonstrated in the general population; this study confirms that the same is true in the workplace. The higher prevalence in women is an established fact in the literature [74]. Conclusive studies have associated bereavement with frequent and severe headaches [75]. In previous studies, the effort/reward imbalance (one of the standard stress models) has been associated with headache [27], while recent studies are beginning to indicate the role of leadership style in the genesis of headache [32]. Factors such as off-time work that interfere with family life have also been associated with headaches in workers [76]. Stress is a nonspecific response of the body to any demand imposed upon it that causes homoeostasis disruption and leads to symptoms such as anxiety, depression, or even headaches [77]. It has been reported that occupational categories such as healthcare workers have a higher rate of migraine than the general population due to heavy workloads, emotional stress, and sleep disturbances related to rotating night shifts [78]. Studies conducted on students have demonstrated that several psychosocial factors (e.g., depression and sleep problems) are significantly associated with headache [79]. In cross-sectional studies, severe insomnia has been associated with headache impact and frequency as measured by the HIT-6 [80]. This investigation confirmed the close association between headaches and sleep problems. 

This study should pave the way for other workplace investigations. Subsequent studies will be able to evaluate which of the various work tasks may be most affected by headache. Longitudinal studies will also determine the type and quantity of the productive problems (e.g., errors, absences from work, accidents and injuries, etc.) that are most associated with headaches.

The significant impact of headache on productivity has prompted companies to develop health promotion interventions. Programs have typically addressed just one of the factors that could induce headaches. The authors have often dealt with muscle tension headaches, which can be diminished by relaxing muscle tension. A multi-component intervention combining workstation ergonomics, group health promotion workshops, neck exercises, and an app for assessing possible reductions in the economic and individual burden of prevalent and incident neck pain and headache in office workers was conducted in Swiss office employees [81]. In another study, the introduction of workplace relaxation exercises significantly decreased pericranial/cervical muscle tenderness and consequently reduced headaches in office workers [82]. Designing residential spaces that integrate light therapy, relaxation opportunities, mindfulness meditation, listening to music, physical activities, aromatherapy, and quality sleep might favor a reduction in the frequency of headaches [83]. A worksite education program for migraine headaches brought improvements in the severity of disease, lost workdays, and work effectiveness [84]. For nurses, self-care training sessions on relaxation techniques resulted in a reduction in muscle pain, sleeplessness, and headaches [85]. Some studies have tried to improve the quality of sleep by administering melatonin, but insufficient evidence has prevented researchers from establishing whether the administration of this substance enhances the quality and quantity of sleep [86]. There is also little evidence that exercise and acupuncture can reduce the intensity and frequency of workers’ headaches and related disabilities [87]. Visual disturbances in office workers might favor the appearance of tension headaches, but there is not enough evidence to indicate that providing computer users with progressive computer glasses would lead to a considerable decrease in headache [88]. Overall, numerous barriers prevent the implementation of effective programs for reducing headache burden in the workplace [89]. Overall, it would seem that prompt detection of headaches affecting work activity, a correct diagnosis of the disorder, the identification of causes and the introduction of prevention and health promotion are still a long way off.

The main limitation of this study is that the screening was conducted in a workplace during routine periodic examinations and therefore within a tight time frame. Consequently, this investigation simply registered the presence of headaches without specifying the type. However, the workers who agreed to participate in the survey were examined by the occupational physician and, if necessary, invited to have tests conducted in National Health Service facilities to obtain an accurate diagnosis of the disorder and undergo treatment. Another limitation concerns the cross-sectional nature of this study, which enabled us to identify the existence of associations but failed to infer causality. Regular periodic health surveillance will nevertheless enable occupational physicians to observe the frequency and impact of headache over time.

## 5. Conclusions

This study, which, to the best of our knowledge, is one of the few conducted in the workplace to assess the frequency and intensity of headache, confirmed that this disorder is very common, especially in female workers, and has a significant impact on productivity. Leadership style influences the impact of headache on productive activities since the disorder increases when the leader interferes in the worker’s personal sphere and does not adequately reward the work done. Sleep problems are closely associated with headache. Given the widespread nature and importance of headache in the workplace, companies should be particularly aware of the need to encourage health promotion activities designed not only to identify, diagnose and treat cases but also to improve sleep hygiene and reduce the occupational stressors responsible for increasing the impact of headache.

## Figures and Tables

**Table 1 ijerph-19-03712-t001:** Association of individual and occupational variables with headache occurrence. Logistic regression analyses.

Variable	Model I—Univariate	Model II—Multivariate
	OR (CI95%)	OR (CI95%)
Sex	1.81 (1.40; 2.35) ***	1.58 (1.18; 2.10) **
Age	1.00 (0.99; 1,01)	-
Night shift	1.06 (0.80; 1.42)	-
Workplace violence	2.25 (1.62; 3.12) ***	1.25 (0.86; 1.82)
Injury	2.09 (1.33; 3.29) ***	1.09 (0.66; 1.82)
Bereavement	2.27 (1.69; 3.04) ***	1.48 (1.06; 2.06) *
Driving accident	1.98 (1.14; 3.41) *	1.37 (0.75; 2.50)
Intrusive leadership	1.16 (1.10; 1,21) ***	1.09 (1.03; 1.15) **
Off-time work	1.08 (1.04; 1,13) ***	1.03 (0.99; 1.08)
Effort	1.16 (1.10; 1.22) ***	1.00 (0.94; 1.07)
Reward	0.91 (0.88; 0.94) ***	0.99 (0.95; 1.04)
Sleep (PSQI)	1.23 (1.18; 1.28) ***	1.16 (1.10; 1.22) ***

* *p* < 0.05; ** *p* < 0.01; *** *p* < 0.001.

**Table 2 ijerph-19-03712-t002:** Stepwise linear regression models assessing the effect of the individual and occupational variables on the impact of headache on work capacity, measured with the HIT-6 questionnaire.

	Model I Standardized Coefficient Beta	Model II Standardized Coefficient Beta	Model III Standardized Coefficient Beta
Sleep quality	0.384 ***	0.398 ***	0.440 ***
Reward	−0.132 **	−0.157 ***	
Intrusive leadership	0.105 *		

* *p* < 0.05; ** *p* < 0.01; *** *p* < 0.001. Variables excluded: workplace violence, injury, bereavement, driving accident, off-time work, effort.

**Table 3 ijerph-19-03712-t003:** Association of headache with anxiety, depression, and metabolic syndrome. Univariate logistic regression analyses.

Outcome	Odds Ratio (OR)	Confidence Interval 95% (CI95%)	*p*
Anxious (anxiety scale ≥ 5)	1.10	1.09; 1.12	<0.001
Depressed (depression scale ≥ 2)	1.09	1.08; 1.11	<0.001
Obesity	1.02	1.01; 1.03	<0.001
Hypertension	1.01	0.99; 1.02	0.221
Low HDL cholesterol	1.03	1.01; 1.04	<0.001
Hypertriglyceridemia	1.01	0.99; 1.03	0.238
Hyperglycemia	1.01	0.99; 1.03	0.232
Metabolic syndrome	1.02	1.00; 1.04	0.018

## Data Availability

Data are deposited in Zenodo. doi:10.5281/zenodo.6127448.

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
