# Peer review of "Headache in the Workplace: Analysis of Factors Influencing Headaches in Terms of Productivity and Health"

_ijerph, 2022, doi:10.3390/ijerph19063712_

Round 1

Reviewer 1 Report

Article: „The impact of headache in the workplace”.

This article deals with the effects of headaches in the workplace. Therefore, we should expect to find out what aspects of work are affected by the headache observed among employees.

Section Results, lines 182-195 - please complete with specific data. It is not enough to write, for example, that the headache depended on gender, not age. The author should add how the data is presented in terms of numbers and percentages. This also applies to the next variables described in these lines (and throughout the manuscript).

The articles are also read by people who are not involved in statistical research, and they would probably like to get more specific data from the text they read. It is recommended that the author adds a table with numerical and percentage data for each variable. At the moment, it is difficult to assess what percentage of, for example, women and men is affected by the headache. This note applies to all variables in the univariate model.

The discussion should be organized. Do lines 253-254 and 260-261 not contradict each other? So is there any research that shows how headaches affect productivity?

At the same time, the aspect of reduced productivity appears in the Discussion section. Unfortunately, there was no reference to the results of own work. There is no doubt that a headache affects sleep quality and vice versa; but these two variables have a measurable impact, for example, on the increase in errors made at work, on the increase in accidents at work. It would be good to cover this aspect both in the research part of the manuscript and in the later section „Discussion”.

Finally, in my opinion, the title does not reflect the content of the manuscript. The author describes the impact of stress at work, violence at work, etc. on the appearance of headaches among employees. But the title indicates that the analysis should be the opposite (as noted above) - that is, whether headaches affect the quality of work, productivity, absence, and finally accidents at work.

Author Response

Reviewer #1

This article deals with the effects of headaches in the workplace. Therefore, we should expect to find out what aspects of work are affected by the headache observed among employees.

Response: I thank the reviewer for this suggestion, and for advice on improving and developing our article. In the future, it may be possible to evaluate the workers of a company who perform similar tasks and understand which, of the different work tasks, have suffered the most from headache. I have included this suggestion among the possible developments of research. In this study, which involved workers with non-homogeneous job tasks, this analysis was not made.

Section Results, lines 182-195 - please complete with specific data. It is not enough to write, for example, that the headache depended on gender, not age. The author should add how the data is presented in terms of numbers and percentages. This also applies to the next variables described in these lines (and throughout the manuscript).

R.: All the data verbally reported in the paragraph are reported in numerical form in the table to which the text refers. I did not find it useful to report the numerical data in the text, because the table is placed immediately after the text. To give an example of how much reading is hindered by the insertion of numbers, I report below the first two sentences mentioned, with the insertion of the numbers that come from the table

“A significantly increased risk of headache was found for female gender (OR= 1.81 CI95%= 1.40; 2.35 p<0.001), while age was not significantly associated with the disorder (OR= 1.00 CI95%= 0.99; 1,01). In univariate models, the risk of headache was significantly higher in individuals who had suffered violence at work (OR=2.25 CI95% 1.62; 3.12 p<0.001), domestic accidents (OR=2.09 CI95%=1.33; 3.29 p<0.001), road accidents (OR= 1.98 CI95%=1.14; 3.41 p<0.05) or bereavement (OR= 2.27 CI95%1.69; 3.04 p<0.001) in the previous year. …”

The articles are also read by people who are not involved in statistical research, and they would probably like to get more specific data from the text they read. It is recommended that the author adds a table with numerical and percentage data for each variable. At the moment, it is difficult to assess what percentage of, for example, women and men is affected by the headache. This note applies to all variables in the univariate model.

R.: The reviewer touched on a very important point. I absolutely agree that the article must also be accessible to people who do not have a thorough knowledge of statistics. Precisely for this reason I have avoided putting all the numerical results in the text and have put them in the tables, as I explained above. However, I believe that the reader of a public health and occupational medicine journal knows what an odds ratio and a confidence interval are and therefore understands that the risk estimate, especially if adjusted for the main confounding factors from a multivariate study, is much more informative than a percentage. The percentage, in fact, is always influenced by the composition of the sample, while the adjusted risk is not. The inferential statistic allows us to progress from the observation of a sample to a knowledge of the universe. In accordance with the reviewer’s suggestion, I added the indication of the number of women and men suffering from headache in this sample.

The discussion should be organized. Do lines 253-254 and 260-261 not contradict each other? So is there any research that shows how headaches affect productivity?

R.: I thank the reviewer who correctly pointed out an apparent contradiction in the references. The studies in lines 253-254 concern the effect of headache on productivity, while the authors cited in lines 260-261 studied the factors that influence this impact. I have changed the text making this difference clearer.

At the same time, the aspect of reduced productivity appears in the Discussion section. Unfortunately, there was no reference to the results of own work. There is no doubt that headache affects sleep quality and vice versa; but these two variables have a measurable impact, for example, on the increase in errors made at work, on the increase in accidents at work. It would be good to cover this aspect both in the research part of the manuscript and in the later section „Discussion”.

R.: I am indebted to the reviewer for pointing out a very important point. I believe that this reviewer's contribution is very useful for planning a multi-year longitudinal study with repeated measurements of the frequency of headache, its impact on work, and sleep quality subjectively measured by workers (as in this cross-sectional study), as well as errors, or injuries, or road accidents or another outcome that could be related to sleep and headache. In this study, I only considered the effect of headache on anxiety, depression, and metabolic syndrome. I took advantage of the suggestion, placing this project among the research developments in the Discussion section.

Finally, in my opinion, the title does not reflect the content of the manuscript. The author describes the impact of stress at work, violence at work, etc. on the appearance of headaches among employees. But the title indicates that the analysis should be the opposite (as noted above) - that is, whether headaches affect the quality of work, productivity, absence, and finally accidents at work.

R.: I agree, the title was too generic. A much more explanatory title could be: “Headache in the workplace. Prevalence and factors associated with the impact on productivity and health”.

Reviewer 2 Report

I have carefully read the article "The impact of headache in the workplace". I think he is well looked after and has a solid studio structure. In particular, the impact of headaches in the workplace is indicated. The research looks interesting and deserves to be published. I believe that some suggestions can be given.
- has the impact of covid 19 on this disorder been evaluated? For example, this article for worker safety could be briefly mentioned doi: 10.3390/healthcare9010017 and doi: https://doi.org/10.1093/intqhc/mzaa085. Brief comments could be indicated in the discussion.
- Could there be suggestions on how to limit the impact of this disorder on workers? is there any research on it?
thank you

Author Response

Reviewer #2

I have carefully read the article "The impact of headache in the workplace". I think he is well looked after and has a solid studio structure. In particular, the impact of headaches in the workplace is indicated. The research looks interesting and deserves to be published. I believe that some suggestions can be given.

- has the impact of covid 19 on this disorder been evaluated? For example, this article for worker safety could be briefly mentioned doi: 10.3390/healthcare9010017 and doi: https://doi.org/10.1093/intqhc/mzaa085. Brief comments could be indicated in the discussion.

Response: I sincerely thank the reviewer for the appreciation of the work and for advice on improving it. The study was concluded prior to the COVID.19 pandemic and therefore did not record the alterations associated with the pandemic. I gladly cited the references indicated by the reviewer along with other studies that refer to the headache phenomenon during COVID-19, even if this problem arose immediately after the conclusion of this study.

- Could there be suggestions on how to limit the impact of this disorder on workers? is there any research on it?

R.: I have discussed the issue of reduction of the burden of headache and have mentioned the initiatives that have been implemented for prevention in some companies.

Reviewer 3 Report

The topic is very interesting and very specific for the contemporary world. The research is well sustained by scientifically sound methods. The statistical analysis is well performed. Overall, this is a well written paper.

Nevertheless, there are some suggestions which I hope will improve the current paper.

Abstract: I suggest to re-structure the abstract according to the recommendations for authors (background, methods, results and conclusion). I also recommend to shorten the information in the results section.

I do not see any key word related to headache and workplace- which would cover the main topic of the research.

Introduction

Substantial information offered in the introduction part, a good background for the research, but updated information are needed.

The first reference sends to a quite old ranking- about 15 years. So if possible and if there are available data, I recommend an up-to-date information about the rankings of headache in disabling people, as well as the percent of people suffering of this disorder (lines 33-36).

Line 47: I suggest to add a citation for micro-pollutants and ergonomic problems.

Line 50: can you specify which occupational environmental factors were related to headache in office workers in the cited study?

Results

Line 187: I think “some stress factors” is not a professional expression. Did you choose which factors to analyze or you analyzed all the items offered by the questionnaire? If you choose, what was the rationale? If you did not, please specify otherwise than “some factors”. I am assuming “some” are listed in lines 189-195.

Discussion

Lines 234-241 need citation.

I do not see discussion of many of the significant results obtained after multivariate logistic regression: female workers, intrusive leadership, bereavement, reward for work and obesity.

Author Response

Reviewer #3

The topic is very interesting and very specific for the contemporary world. The research is well sustained by scientifically sound methods. The statistical analysis is well performed. Overall, this is a well written paper.

Nevertheless, there are some suggestions which I hope will improve the current paper.

Response: I sincerely thank you for the appreciation expressed towards the work and for advice on improving it.

Abstract: I suggest to re-structure the abstract according to the recommendations for authors (background, methods, results and conclusion). I also recommend to shorten the information in the results section.

Response: I have tried to adhere to the editorial guidelines, which state that the abstract should be a single paragraph and should follow the style of structured abstracts, but without headings; it should be a total of about 200 words. Adhering to the reviewer's invitation I have therefore reduced the length of the abstract by deleting part of the results.

I do not see any key word related to headache and workplace- which would cover the main topic of the research.

Response: In accordance with a scholastic indication, I avoided repeating words in the keywords that had already been used in the title, because both the title and the keywords contribute to the bibliographic classification of the article. I have added the term “headache disorders” and the term “workplace” health promotion.

Introduction

Substantial information offered in the introduction part, a good background for the research, but updated information are needed. The first reference sends to a quite old ranking- about 15 years. So if possible and if there are available data, I recommend an up-to-date information about the rankings of headache in disabling people, as well as the percent of people suffering of this disorder (lines 33-36).

R.: I gladly accept the reviewer's request and have updated the indications. I have removed the outdated  reference and put three very recent references, two of which were already present in the previous version.

Line 47: I suggest to add a citation for micro-pollutants and ergonomic problems.

R.: I have gladly added two recent references,

Line 50: can you specify which occupational environmental factors were related to headache in office workers in the cited study?

R.: I have mentioned the main factors to which the people interviewed attributed the problem.

Results

Line 187: I think “some stress factors” is not a professional expression. Did you choose which factors to analyze or you analyzed all the items offered by the questionnaire? If you choose, what was the rationale? If you did not, please specify otherwise than “some factors”. I am assuming “some” are listed in lines 189-195.

R.: in line 187 I have indicated more correctly "the most common occupational stress factors". I also explained in the Methods the criteria for choosing these occupational stress factors. Both common occupational trauma and intrusive leadership and off-time work have been associated with workers' mental health in previous studies.

Discussion

Lines 234-241 need citation.

R.: The first paragraph of the Discussion contains, as usual, the results of this study. To make this fact clearer, I have prefaced the sentence "This study has shown that…"

I do not see discussion of many of the significant results obtained after multivariate logistic regression: female workers, intrusive leadership, bereavement, reward for work and obesity.

R.: I am very pleased with this observation because it allowed me to add some hints on aspects of this study that should be emphasized and compared with the literature.

Reviewer 4 Report

Dear author,

Thank you for a great study of importance to understand underlying conditions and circumstances of workplace wellbeing. I have actually very few suggestions to make on this manuscript, since it is well written and concise.

#1. In what branch (typ of work) was this study conducted? Please add in the method section.

#2. Since I get the impression that this is a study published by one author, I suggest revising the stirring of “we” in the discussion and conclusion section.

Otherwise, I congratulate to emphasis a common symptom and adding scientific knowledge on this phenomena.

Author Response

Reviewer #4

Dear author,

Thank you for a great study of importance to understand underlying conditions and circumstances of workplace wellbeing. I have actually very few suggestions to make on this manuscript, since it is well written and concise.

R.: I sincerely thank my colleague for the appreciation and advice he/she has expressed.

#1. In what branch (typ of work) was this study conducted? Please add in the method section.

R.: I have gladly added this indication in the description of the convenience population. The activities were heterogeneous and included: office work, industrial (plumbing, electricity), trade (fuel distribution, food), services (personal assistance, cleaning)

#2. Since I get the impression that this is a study published by one author, I suggest revising the stirring of “we” in the discussion and conclusion section.

R.: I have promptly corrected all parts of the manuscript in which there was the term "our"

Otherwise, I congratulate to emphasis a common symptom and adding scientific knowledge on this phenomena.

R.: I sincerely thank you

Reviewer 5 Report

The paper discuss about interesting issue that is headache. I think authors should adjust the title of the article to the conducted research. In my opinion, the title is too general and does not follow from it. The title should attract and encourage the reader. Therefore, I propose to refine the title. 

In the introduction, the reasons why the authors chose such an analysis are missing. What is the cause. Literature analysis is also very poor. When taking up the topic of headache in the workplace, they expect statistical data about the situation in the selected country in the light of selected countries. How do I deal with the problem, employees and companies, how do they support employees so that they do not have stress and therefore a headache. Therefore, I propose to refine this part, especially on the basis of the latest studies. In 2.2. Questionnaire point - lack information about who were respondents  - how many of them, how were they creating base on gender, age end so on.  How the respondents were selected. Were they matched for headache sufferers into the 3 categories mentioned by the authors?

Too short conclusion. Authors write that: "Given the widespread nature and importance of headache in the workplace, companies should have a great interest in encouraging health promotion activities that in addition to identifying... " from my knowledge I know that companies undertake a lot of such initiatives. So in my opinion you should present them just here. 

There are a few recent studies in the literature review. There are grammar errors in the work. The work requires careful linguistic correctness.

Author Response

Reviewer #5

The paper discuss about interesting issue that is headache. I think authors should adjust the title of the article to the conducted research. In my opinion, the title is too general and does not follow from it. The title should attract and encourage the reader. Therefore, I propose to refine the title.

R.: I thank the reviewer who appreciated the work and made very helpful comments for improving it. I have accepted the suggestion about making the title more specific and descriptive.

In the introduction, the reasons why the authors chose such an analysis are missing. What is the cause. Literature analysis is also very poor. When taking up the topic of headache in the workplace, they expect statistical data about the situation in the selected country in the light of selected countries. How do I deal with the problem, employees and companies, how do they support employees so that they do not have stress and therefore a headache. Therefore, I propose to refine this part, especially on the basis of the latest studies.

Response: Gladly accepting the reviewer's invitation I have rewritten the Introduction, explaining better than I did in the first version that headache is one of the non-communicable diseases that causes the greatest loss of years of health and how international bodies are actively engaged in reducing the burden of headache. I underlined in the first paragraph the reference to the Global Burden of Disease study from which these data are taken, and I cited an article that contains comparisons between countries of the world.

I then explained that because of its importance, headache was the object of a health promotion action in some small companies and that this article was written with data obtained from health surveillance in the workplace.

I therefore dealt with the health promotion experiences conducted in other companies in the Discussion.

In 2.2. Questionnaire point - lack information about who were respondents  - how many of them, how were they creating base on gender, age end so on.  How the respondents were selected. Were they matched for headache sufferers into the 3 categories mentioned by the authors?

R.: I explained that the survey was based on a convenience sample consisting of small companies whose workers are subjected to health surveillance by the author. In this version of the manuscript I have detailed, in addition to the sector to which they belong, also the type of job. The jobs were very heterogeneous and there was no need for matching by company, job sector or type of job, because the invitation to participate was addressed to everyone and participation was very high (97%).

Too short conclusion. Authors write that: "Given the widespread nature and importance of headache in the workplace, companies should have a great interest in encouraging health promotion activities that in addition to identifying... " from my knowledge I know that companies undertake a lot of such initiatives. So in my opinion you should present them just here.

Response: In the previous version, lines 272-292 of the Discussion were dedicated to headache prevention interventions in companies. In this version I have added two studies that I have heard of. I did not find any other studies of workplace health promotion for headache in the literature.

There are a few recent studies in the literature review. There are grammar errors in the work. The work requires careful linguistic correctness.

R.: I made sure that the most recent citations of studies on headache in the workplace were in the bibliography. Because I am not a native speaker of English, before sending the first version of the article I arranged a language review by a native speaker with over 20 years of experience in language proofing. I am quite sure there was no grammar error in the text.

Round 2

Reviewer 1 Report

Thank you to the author for clear explanations.

Author Response

Reviewer #1

Thank you to the author for clear explanations.

Reviewer #2

I thank the authors who followed the instructions and improved the text. I consider the article to be published

Reviewer #5

Thank you for improving the paper. I propose changing the title to "Analysis of factors influencing headaches in the workplace in terms of productivity and health."

Response: I gladly accept the advice of reviewer # 5 regarding the title of the manuscript.

I thank all the reviewers for the commitment and attention with which they have evaluated my work

Reviewer 2 Report

I thank the authors who followed the instructions and improved the text. I consider the article to be published

Author Response

(The authors gave the same response as above.)

Reviewer 5 Report

Thank you for improving the paper. I propose changing the title to "Analysis of factors influencing headaches in the workplace in terms of productivity and health."

Author Response

(The authors gave the same response as above.)
